# Dental Caries, Tooth Loss and Quality of Life of Individuals Exposed to Social Risk Factors in Northeast Brazil

**DOI:** 10.3390/ijerph20176661

**Published:** 2023-08-28

**Authors:** Luísa Simões de Albuquerque, Raiana Gurgel de Queiroz, Jenny Abanto, Marcelo José Strazzeri Bönecker, Franklin Delano Soares Forte, Fábio Correia Sampaio

**Affiliations:** 1Department of Clinical and Community Dentistry, Health Science Center, Federal University of Paraíba, João Pessoa 58051-900, PB, Brazil; luisasimoesa@gmail.com (L.S.d.A.); raianagqueiroz@gmail.com (R.G.d.Q.); franklinufpb@gmail.com (F.D.S.F.); 2Internacional University of Catalunya, Catalunya, 08017 Barcelona, Spain; jennyabantousp@gmail.com; 3Faculty of Dentistry, University of São Paulo, São Paulo 05508-220, SP, Brazil; bonecker@usp.br

**Keywords:** dental health survey, dental caries, tooth loss, edentulous, socioeconomic factors, oral health-related quality of life

## Abstract

This study aimed to investigate the impact of dental caries and tooth loss on oral health-related quality of life (OHRQoL) in socioeconomically disadvantaged people. A population-based, cross-sectional survey was conducted in 28 cities at social risk in Northeast Brazil. All permanent residents aged 12, 15–19, 35–44, and 65–74 years were eligible, and 3063 were included. Participants answered a questionnaire on socioeconomic status, beliefs, and behaviors. Trained local dentists performed oral clinical examinations during home visits. Caries and tooth loss were evaluated using the decayed, missing, and filled teeth (DMFT) index and OHRQoL was evaluated using the Oral Health Impact Profile-14 (OHIP-14). Poisson regression analysis was performed to assess the relationship between individual domains, OHIP-14 scores, dental caries, tooth loss, and socioeconomic/demographic characteristics. Mean DMFT (standard deviation) scores were 2.68 (4.01), 4.84 (4.30), 15.35 (7.26), and 26.72 (8.03) for groups aged 12, 15–19, 35–44, and 65–74 years, respectively. Most participants (70%) were partially edentulous and 13% were completely edentulous. Caries and tooth loss significantly increased with age and impacted OHRQoL. Physical pain (5.8%) and psychological discomfort (5.8%) were the most commonly reported on the OHIP-14. Untreated caries (prevalence ratio (PR), 1.54; 95% confidence interval (CI), 1.37–1.72) and edentulism (PR, 1.29; 95% CI, 1.08–1.53) had a significant negative impact on OHRQoL. Income, level of education, sex, age, and oral hygiene habits were also related to OHRQoL. There was a high prevalence of dental caries and edentulism in all age groups except 12-year-olds. OHRQoL was negatively impacted by these oral conditions across the lifespan, with a trend towards more negative scores and higher impact in older adults.

## 1. Introduction

Untreated carious lesions in permanent teeth was the most prevalent health condition globally in 2010, affecting more than two billion people or approximately 35% of the world’s population [1]. Although the prevalence of untreated dental caries has decreased by 4% in the last decade, a significant number of individuals worldwide are still affected, and the burden associated with dental caries is likely to impact their quality of life [2,3]. 

Oral health-related quality of life (OHRQoL) is a multidimensional construct that includes a subjective evaluation of the individual’s oral health. It is regarded as a good indicator of the impact that oral health and disease have on an individual’s daily functioning, well-being, and overall satisfaction with daily life. Recent studies report that poor oral health significantly affects both functional and social activity, with a consequent reduction in quality of life [4,5,6,7]. There is evidence that a high burden of decayed teeth significantly affects quality of life for people in Latin American and Caribbean countries [8,9]. 

Among Brazilians, caries in primary and permanent dentition is characterized by early onset, a high prevalence of untreated caries, and increasing severity with age [10,11,12]. Caries prevalence is heterogeneous across different regions of the country. In Northeast Brazil, for example, the prevalence of dental caries and tooth loss (edentulism) is moderate to high; children 12 years of age living in the countryside of this region have an average decayed, missing, and filled teeth (DMFT) index score significantly higher than the national average (3.84 vs. 2.1) [10]. In addition, the prevalence of caries is set to increase in the coming years due to the growing aging population and a concomitant reduction in levels of edentulousness [5,13]. 

Dental caries is unevenly distributed among the Brazilian population, with women and those with the lowest incomes disproportionately affected [14]. Edentulism is highly prevalent in Brazil and is linked to social inequality and inequity. Based on the most recent national epidemiological survey, 92.7% of individuals aged between 65 and 74 years were completely or partially edentulous, with a greater number of affected individuals living in the countryside in the North and Northeast regions of Brazil [10]. 

It is important to note that there are few epidemiological data on the oral health of people in the Brazilian countryside and the available information is outdated. Furthermore, there has been little investigation into the potential impact of dental caries and edentulism on OHRQoL in populations with socioeconomic disadvantages [15,16,17]. In addition, most studies examining dental caries and OHRQoL in residents of Brazil have focused on specific age groups, such as children or adolescents, or individuals with specific needs or conditions [18,19,20]. There are few studies on OHRQoL in the Northeast region of the country. In 2013, an ecological study indicated that socio-economic disadvantages in the region impact heavily on this population’s self-perception of oral health [21]. A recent study showed that the most affected OHIP-14 dimensions in older adults using prothesis were physical pain in the oral region, psychological discomfort, psychological disability, and physical disability [22]. However, this study did not cover the countryside area where access to preventive care is a major problem. Surveying OHRQoL in this area can add a powerful dimension in the planning and development of oral health promotion programs.

The aim of this cross-sectional, population-based study was to investigate the impact of dental caries, tooth loss, and social determinants on the OHRQoL of individuals across a wide age range with socioeconomic disadvantages associated with poor health outcomes.

## 2. Materials and Methods

### 2.1. Study Design

This was a population-based, cross-sectional study carried out in accordance with STROBE guidelines [21,22,23] between April 2015 and April 2017. Participants were examined to assess their oral health and then completed the Oral Health Impact Profile-14 (OHIP-14) questionnaire to evaluate their OHRQoL.

### 2.2. Ethics

The study protocol was approved by the Ethics Committee for Research Involving Human Beings of the Federal University of Paraíba (CAAE: 3087414.9.0000.5188), followed guidelines and regulations for research involving humans, and complied with resolution 466/2012 of the National Council Ministry of Health, Brasília—DF. All participants in the study provided informed consent.

### 2.3. Study Location

Paraíba is a low-to-middle-income state with a population of 4 million. It was ranked in the bottom five Brazilian states in terms of the 2010 Human Development Index (HDI), with an HDI score of 0.658 [22,23,24]. In 2020, the per capita income in Paraíba was approximately US$307, ranking it among the lowest-income regions in Brazil (IBGE, 2020). Paraíba has the seventh lowest proportion of people with health insurance in the country and according to the National Health Survey, only 12.2% of the population has access to health services [22].

The 28 municipalities included in this study were classified as areas of social vulnerability by the Ministry of Health. These areas met the following inclusion criteria: (1) city (urban areas only) with less than 50,000 inhabitants; (2) lower HDI according to indicators in the Atlas of Human Development in Brazil 2013 [23,24,25]; (3) listed as a priority municipality by the Federal Government’s “*Brasil Sem Miséria*” Plan, which includes localities for social interventions for human development [24,25,26,27]. The HDIs of the included cities ranged from 0.513 in Gado Bravo (8365 inhabitants) to 0.628 in Serra Branca (12,973 inhabitants). 

### 2.4. Study Population and Data Collection

Our approach followed the same sampling plan used by the National Oral Health Survey [28]. Briefly, all 28 cities selected have up to 6 enumeration areas, which is a digital map that corresponds to the smallest territorial area used by the Brazilian Institute of Geography and Statistics (IBGE). These enumeration areas depict all residential houses as well as social facilities within a specific area of a city [29]. Essentially, each of the municipalities selected for the study comprised a domain (urban area). In twenty cities (5000–15,000 inhabitants), two or three enumeration areas covered the whole domain. Only eight cities (16,000–20,000 inhabitants) have five or six enumeration areas. In these domains, a simple random selection of the enumeration was carried out. This was conducted to investigate a maximum of four enumeration areas per domain (city). Finally, a total of 77 enumeration areas were included in the study. Based on the Brazilian census [30], it was estimated that there was a total of 19,500 inhabitants in the municipality. However, part of this population lived in rural communities and, therefore, was excluded. After considering only the relevant age groups in the urban areas, the estimated total target population was approximately 12,000 inhabitants. 

All participants (in the target age range) that were living in the enumeration areas (urban zone) were invited to participate in the study using a census method. A list of all addresses and permanent residents within the target age range living in each enumeration area was prepared. Examiners made home health visits during which an oral examination was carried out. If the examination and data collection could not be performed after three consecutive attempts, the house and consequently the participants were excluded from the study. 

### 2.5. Oral Clinical Examination

Field teams were composed of dentists (n = 42), dental auxiliaries (n = 42), and community health agents (n = 64). Teams were trained and techniques standardized through annual, in-person workshops for 8 or 10 field teams that included an examiner and a note-taker. Intra- and interexaminer reliability was calculated using the Kappa agreement test (for dental caries) weighted for each examiner, age group, and condition studied, with a value of 0.65 as the minimum acceptable limit. Intra-examiner reproducibility was assessed by comparing results obtained by examiners on the first day of analysis with those obtained after one week of examinations. In the field, all visits began with supervised brushing of teeth followed by oral examination under indirect natural light using probes as recommended by the World Health Organization (WHO) [28]. The DMFT index was used to evaluate caries and tooth loss [31]. 

Dental health indicators (predictors) were derived from individual tooth- and surface-level data and were used to calculate the number of decayed surfaces (D), number of decayed, missing, filled surfaces (DMFS) index, and significant caries (SiC) index. 

### 2.6. General Questionnaire

All participants examined were invited to answer a questionnaire on their socioeconomic status, use of dental services, health habits, and oral health self-perception. Participants were also questioned about their social characteristics, including family structure, education, household overcrowding, income, participation in social activities, self-perception as a religious individual, tobacco use, and alcohol consumption. Information on the family and minors was obtained from the head of the household; individuals who had reached the Brazilian legal age of 18 years when examined could answer for themselves.

### 2.7. OHRQoL

This study used the Portuguese version of the OHIP-14 [32]. It is worth noting that a translated, adapted, and validated version of OHIP-14 has been available for Brazilians since 2003 [33,34,35]. Psychometric evaluation of the OHIP-14 was also carried out in Brazil among adults [36] and adolescents living in rural areas [36,37,38]. 

OHIP-14 total score was calculated by summing responses over all 14 items. This score ranges from 0 to 56, with a higher score indicating a greater negative impact on OHRQoL. Based on a previous study, the total and final scores in each domain were combined to form a binary result of either ‘impact present’ (for “often” or “all the time” answers) or ‘no impact’ (for “never”, “hardly ever”, or “sometimes” answers) (Mussolini, 2020). The total scores were also classified into three levels based on percentiles: [0–8) (P75, indicating the 75th percentile), [8–15) (P75–90, indicating the 75th–90th percentile range, and ≥15 (>P90, for scores above the 90th percentile) [37,38].

### 2.8. Statistical Analysis

Statistical analyses were performed using SPSS 22 (IBM Corp, Armonk, NY, USA) and STATA 8.0 (Stata Corp, College Station, TX, USA). Data were stratified according to sex, age, and age group. The Kolmogorov–Smirnov test was used to evaluate normality. Due to non-normal distribution, comparisons were carried out nonparametrically, using a Kruskal–Wallis test followed by a post-hoc Bonferroni-adjusted Mann–Whitney test. Bivariate analyses were performed with chi-square and Fisher’s exact tests. Correlations between OHIP-14 and DMFT and its components were analyzed using a Spearman correlation test. Poisson regression analysis with robust variance was performed to correlate individual domains and total OHIP-14 scores with dental caries, tooth loss, and socioeconomic/demographic characteristics of participants [39]. Poisson regression analysis was performed to select variables with a *p*-value ≤ 0.20 to enter the model. Then, explanatory variables selected were tested in the multivariate model and retained only if they had a *p*-value of ≤ 0.05. This analysis employed a count outcome, and prevalence ratios (PRs) and 95% confidence intervals (95% CI) were calculated.

## 3. Results

A total of 4076 individuals were invited to participate. Of these, 498 refused the dental examination and 515 refused to respond to the OHIP-14 questionnaire. As a result, the final number of individuals included in this survey was 3063. Of these, 194 were 12 years old, 817 were 15–19, 1302 were 35–44, and 750 were 65–74 years of age. 

Mean DMFT (SD) scores were 2.68 (4.01), 4.84 (4.30), 15.35 (7.26), and 26.72 (8.03) in groups aged 12, 15–19, 35–44, and 65–74 years, respectively. All mean DMFT scores were higher than national mean scores in Brazil. Seventy percent of participants were partially edentulous and 13% were completely edentulous. Caries prevalence (DMFT and each component) and tooth loss significantly increased with age and had a negative impact on OHRQoL. Untreated caries (prevalence ratio (PR), 1.54; 95% confidence interval (CI), 1.37–1.72) and edentulism (PR, 1.29; 95% CI, 1.08–1.53) had a significant negative impact on OHRQoL.

Table 1 shows the relationship of the sociodemographic characteristics of the participants with the OHIP-14 classified in three percentiles (P75, P75–P90 and >P90). OHIP-14 was virtually related to all characteristics investigated except for household agglomeration, having their own toothbrush, smoking and alcohol consumption. 

The mean scores for DMFT index and its components are presented in Table 2 along with OHIP-14 scores according to age. A clear trend towards higher OHIP-14 scores with increasing age was observed. This trend was also seen for all other components, except decayed teeth (DT) and filled teeth (FT) in the group aged 65–74 years. This is likely due to the high level of missing teeth in this age group. 

Appendix A shows the distribution of individuals according to OHIP-14 dimensions. Considering the potential impact in quality of life indicated by the association of the most negative answers (“often” and “all the time”), it could be observed that physical pain (5.8%) and psychological discomfort (5.8%) were the most frequently occurring OHIP-14 dimensions (Table 2).

Univariate analysis showed that recent topical fluoride application, need for a dental prosthesis, prevalence of edentulism, tooth loss due to caries, prevalence of caries experience, and prevalence of untreated caries were all correlated with OHRQoL (*p* < 0.05) (Table 3). Religious belief had a positive impact on the individuals’ quality of life (relative risk (RR), 1.02; 95% CI, 0.86, 1.21; *p* = 0.831). 

The final adjusted multivariate model included five covariates (Table 4). Increasing prevalence of untreated caries had a negative impact on the participants’ OHRQoL (RR, 1.54; 95% CI, 1.37, 1.72; *p* < 0.001). Also, a family income of at least two Brazilian minimum wages (BMW) had a positive impact on the parents’ OHRQoL (PR, 1.06; 95% CI, 0.99, 1.20; *p* = 0.323) (Table 4). All analyses had a power of >90%.

## 4. Discussion

This survey investigated the impact of experience with dental caries and tooth loss (partial and complete edentulism) on the OHRQoL of Brazilians who were socioeconomically disadvantaged. To the best of our knowledge, this is the first population-based survey applying a census method to evaluate this relationship in >3000 inhabitants of a challenging environment. While a cross-sectional study design has known limitations in the interpretation of statistical analyses, our survey evaluated experience with oral health and its impact on quality of life across different age groups, providing useful information about their interrelatedness across the lifespan. Importantly, the study assessed dental caries after participants had brushed their teeth, allowing examiners to detect caries more easily on clean tooth surfaces. 

The cities included in this study share similar social and environmental challenges that impact the local economy, education, and many other aspects of the lives of participants. For example, the Paraíba state countryside—particularly the semiarid region—had little rainfall between 1981 and 2019. As a result, many participants in this study had difficulty obtaining a regular supply of potable water. In addition, some participants were also faced with food insecurity [40].

Twenty years ago, at least 30% of the population of Paraíba was vulnerable, in other words, unable to afford a basic daily caloric intake. Nevertheless, quality of life as measured by an index encompassing 21 social indicators increased by 37% between 1990 and 2000, the largest improvement by a single state in Brazil during that decade [41]. However, this positive trend is not likely to have continued due to the 2020–2021 COVID-19 pandemic and because the social vulnerability of small cities in Brazil has been reported to be increasing [42]. Thus, our data may provide a useful pre-pandemic baseline for future surveys. 

This study clearly indicates that caries and edentulism have a substantial impact on the quality of life of this population. All DMFT scores as well as scores for its components were higher than national averages. Interestingly, in the 12-year-old age group, DT scores were the highest of all components measured, whereas FT was highest in those aged 15–19 years. Missing teeth were the most important component for the groups aged 35–44 and 65–74 years. These data illustrate the impact of dental caries across the lifespan, ultimately leading to high rates of edentulism as individuals age. 

DMFT scores showed that very few participants had a DMFT score of zero, meaning they could be classified as caries-free. In contrast, OHIP-14 scores were very much skewed to the left, with median scores of zero observed for all age groups except those aged 35–44 years (Table 2). 

Confirming Oliveira’s study (2021) [43], women in this survey reported that their experience of dental caries and untreated dental decay had a greater impact on OHRQoL, with a mean OHIP-14 score of 3.43 (95% CI 2.80 to 4.21) (Table 3).

Some of the variables reported in Table 3 that are related to OHIP-14 scores may be confounding. For example, individuals with more than 10 years of education could have more experience with caries, and respondents with an income greater than 3 BMW may report a greater negative impact on OHRQoL. This phenomenon has been reported previously [39,44]. In a study carried out by [45,46,47], several comparisons performed using complex regression analyses became confounding factors. As a result, interpreting this type of analysis must be performed cautiously, since some relationships are non-linear, and there may be variables that are not examined but that affect the outcome.

Some limitations of this study must be addressed. The use of OHIP-14 among adolescents can be questioned since data from children and young adults present unique challenges. For instance, the children’s dental, facial, and cognitive development changes drastically throughout childhood and adolescence, and other indexes (e.g., OIDP) could be used for these age groups [48]. Comparisons with local studies are sometimes difficult since most studies investigate OHRQoL on specific age groups with different health situations [22]. The diversity of modes of presentation of data in OHRQoL is also a problem.

Finally, it is important to note that participants in this study had limited access to private oral health services. Access to oral health care in these cities is mainly provided by the Primary Health Care (PHC) from The Brazilian National Health System (SUS). The high prevalence of untreated caries along with a finding of few filled teeth suggests that PHC services in this region are fragmented. Nevertheless, evaluating OHRQoL remains challenging because the individual’s psychosocial perception of their oral health condition can be complex. This suggests that individuals in this population may not have fully appreciated or acknowledged the problems they faced [49,50]. Thus, providing a dental examination at home with a carefully supervised oral hygiene session as data were collected during this study may have reinforced the participants’ impression that their treatment needs were being met. However, this impression was not shared by edentulous participants. Our survey showed that edentulism at an early age had a long-lasting psychological impact on study participants, and that the psychological discomfort outweighed any functional limitation. Effective public policies that include provision of access to appropriate dental care that can control dental caries and consequently reduce edentulism are urgently needed in this region.

## 5. Conclusions

A high prevalence of dental caries and edentulism was found in all age groups except 12-year-olds in this population. OHRQoL was negatively impacted by these conditions in all target age groups surveyed. There was a trend towards more negative OHIP-14 scores in older adults, suggesting that the effects of dental caries are cumulative and that OHRQoL is negatively influenced across the lifespan.

## Figures and Tables

**Table 1 ijerph-20-06661-t001:** Socioeconomic profile and studied variables according to Oral Health Impact Profile (OHIP-14) in individuals from Paraíba, Brazil.

Sociodemographic Characteristics	Total	OHIP-14			
		[0–8.0)	[8.0–15.0)	≥15.0 **	*p* *
	n (%)	n (%)	n (%)	n (%)	
Sex					0.011
*Male*	1188 (38.8)	956 (80.5)	171 (14.4)	61 (5.1)	
*Female*	1875 (61.2)	1424 (75.9)	322 (17.2)	129 (6.9)	
Income					
*≤2 minimum wages (BMW)*	1882 (61.4)	1477 (78.5)	283 (15.0)	122 (6.5)	0.031
*≥3 minimum wages (BMW)*	922 (30.1)	691 (74.9)	177 (19.2)	54 (5.9)	
*No reply*	259 (8.5)	212 (81.9)	33 (12.7)	14 (5.4)	
Household agglomeration					
*Ideal (* *≤2 person per bedroom)*	2685 (87.7)	2088 (77.8)	437 (16.3)	160 (6.0)	0.077
*Not ideal (>2 person per bedroom)*	277 (9.0)	207 (74.7)	43 (15.5)	27 (9.7)	
*No reply*	101 (3.30)	85 (84.2)	13 (12.9)	3 (3.0)	
Spiritual person					
*Yes*	2270 (74.1)	1780 (78.4)	345 (15.2)	145 (6.4)	0.009
*No*	677 (22.1)	501 (74.0)	133 (19.6)	43 (6.4)	
*No reply*	116 (3.8)	99 (85.3)	15 (12.9)	2 (1.7)	
Own toothbrush					
*Yes*	2953 (96.4)	2298 (77.8)	469 (15.9)	186 (6.3)	0.088
*No*	45 (1.5)	29 (64.4)	13 (28.9)	3 (6.7)	
*No reply*	65 (2.1)	53 (81.5)	11 (16.9)	1 (1.5)	
Use fluoride of toothpaste					
*Yes*	2887 (94.3)	2251 (78.0)	458 (15.9)	178 (6.2)	0.007
*No*	112 (3.6)	73 (65.2)	30 (26.8)	9 (8.0)	
*No reply*	64 (2.1)	56 (87.5)	5 (7.8)	3 (4.7)	
Brush teeth frequently					
*Yes*	2791 (91.1)	2187 (78.4)	432 (15.5)	172 (6.2)	0.007
*No*	191 (6.3)	128 (67.0)	49 (25.7)	14 (7.3)	
*No reply*	81 (2.6)	65 (80.2)	12 (14.8)	4 (4.9)	
Tobacco use					
*Yes*	317 (10.3)	233 (73.5)	53 (16.7)	31 (9.8)	0.059
*No*	2669 (87.2)	2086 (78.2)	426 (16.0)	157 (5.9)	
*No reply*	77 (2.5)	61 (79.2)	14 (18.2)	2 (2.6)	
Consume alcohol					
*Yes*	644 (21.0)	485 (75.3)	113 (17.5)	46 (7.1)	
*No*	2370 (74.3)	1852 (78.1)	374 (15.8)	144 (6.1)	0.143
*No reply*	144 (4.7)	43 (87.8)	6 (12.2)	0 (0.0)	
Need for prosthesis					
*Yes*	1393 (45.4)	992 (71.2)	268 (19.2)	133 (9.5)	
*No*	1425 (46.5)	1176 (82.5)	199 (14.0)	50 (3.5)	<0.001
*No reply*	245 (8.1)	212 (86.5)	26 (10.6)	7 (2.9)	
Prevalence of caries experience					
*Yes*	2859 (93.3)	2196 (76.8)	477 (16,7)	186 (6.5)	<0.001
*No*	204 (6.7)	184 (90.2)	16 (7.8)	4 (2.0)	
Prevalence of untreated caries					
*Yes*	1497 (48.9)	1221 (81.6)	215 (14.4)	61 (4.1)	<0.001
*No*	1566 (51.1)	1159 (74.0)	278 (17.8)	129 (8.2)	
Prevalence of edentulism					
*Yes*	2200 (71.8)	1645 (74.8)	389 (17.7)	166 (7.5)	<0.001
*No*	863 (28.2)	735 (85.2)	104 (12.1)	24 (2.8)	

* *p* value calculated by chi-square test of Fisher’s exact test. ** Cut-off points selected according to quartiles. Scores [0–8) indicates the values below the 75th percentile (P75), scores [8.0–15.0) = P75–P90 and scores ≥15.0 represents the >P90. BMW = Brazilian Minimum Wage.

**Table 2 ijerph-20-06661-t002:** Distribution of mean (SD), confidence interval and median of DMFT and its components and OHIP-14 total scores, the frequency of answers of “impact present” given by OHIP-14 scores (P90), the number and percentage of “caries-free” and edentulism according to different age groups (n = 3063).

Categories	Age Groups			
	12 Years-Old	15–19 Years-Old	35–44 Years-Old	65–74 Years-Old
Caries experience				
DMFT				
*Mean (SD)*	2.68 (4.01) ^a^	4.84 (4.30) ^b^	15.35 (7.26) ^c^	26.72 (8.03) ^d^
*Confidence interval*	2.11–3.25	4.55–5.14	14.96–15.75	26.14–27.30
*Median*	2.00	4.00	15.00	32.00
Decayed Teeth (DT)				
*Mean (SD)*	1.16 (1.17) ^a^	1.84 (2.53) ^a,b^	2.33 (3.39) ^b^	1.11 (2.66) ^c^
*Confidence interval*	0.92–1.40	1.66–2.01	2.15–2.52	0.92–1.30
*Median*	0.50	1.00	1.00	0.00
Filled Teeth (FT)				
*Mean (SD)*	0.93 (1.70) ^b^	2.16 (3.01) ^c^	4.15 (4.46) ^d^	0.40 (1.51) ^a^
*Confidence interval*	0.69–1.18	1.95–2.36	3.91–4.39	0.29–0.51
*Median*	0.00	1.00	3.00	0.00
Missing teeth (MT)				
*Mean (SD)*	0.58 (3.30) ^a^	0.84 (1.95) ^b^	8.88 (7.72) ^c^	25.21 (9.36) ^d^
*Confidence interval*	0.11–1.05	0.70–0.97	8.46–9.30	24.53–25.88
*Median*	0.00	0.00	7.00	31.00
“Caries-free” n (%)	5 (2.60)	122 (14.90)	14 (1.10)	10 (1.30)
Partial edentulism n (%)	32 (16.50)	276 (33.80)	1,168 (89.70)	724 (96.50)
Total edentulism n (%)	2 (1.03)	3 (0.36)	34 (2.61)	372 (49.6)
OHIP-14				
*Mean (SD)*	2.29 (5.74) ^a^	3.69 (6.34) ^b^	5.91 (8.45) ^c^	4.95 (7.61) ^d^
*Confidence interval*	1.48–3.10	3.25–4.12	5.45–6.37	4.41–5.50
*Median*	0.00	0.00	2.00	0.00
*Impact present n (%)*	2 (1.00)	30 (3.70)	114 (8.80)	44 (5.90)

Different letters indicate statistically significant difference at *p* < 0.05 using Kruskal–Wallis test, followed by Bonferroni-adjusted Mann–Whitney test as post-hoc test. SD, standard deviation.

**Table 3 ijerph-20-06661-t003:** Univariate analysis for association between dental caries experience and untreated dental decay in relation to overall OHRQoL in subjects in Paraíba, Brazil.

Covariates	*n* (%)	Robust RR (95% IC)	*p*-Value *
Age Group			
*12 years-old*	194 (6.3)	1.61 (1.11, 2.34)	0.036
*15–19 years-old*	817 (26.7)	2.58 (1.80, 3.70)	<0.001
*35–44 years-old*	1302 (42.5)	2.16 (1.50, 3.13)	0.002
*65–74 years-old*	750 (24.5)	2.29 (1.61, 3.25)	0.001
Sex			
*Male*	1188 (38.8)	1.24 (1.10, 1.39)	<0.001
*Female*	1875 (61.2)	3.43 (2.80, 4.21)	<0.001
Scholarity			
*Illiterate*	386 (12.6)	0.91 (0.77, 1.08)	0.299
*≤10 of study*	2064 (67.4)	0.94 (0.73, 1.20)	0.618
*>10 of study*	218 (7.1)	5.37 (4.60, 6.27)	<0.001
Household agglomeration			
*Ideal (≤2 person per bedroom)*	2685 (87.6)	1.19 (0.99, 1.43)	0.063
*Not ideal (>2 person per bedroom)*	277 (9.0)	4.03 (3.26, 4.98)	<0.001
Income			
*≤2 minimum wage (BMW)*	1882 (61.4)	1.06 (0.94, 1.20)	0.323
*≥3 minimum wage (BMW)*	922 (30.1)	4.52 (3.79, 5.38)	<0.001
Own toothbrush			
*Yes*	2953 (96.4)	1.25 (0.84, 1.87)	0.273
*No*	45 (1.5)	3.88 (2.57, 5.87)	<0.001
Use fluoride of toothpaste			
*Yes*	2887 (94.2)	1.39 (1.09, 1.77)	0.008
*No*	112 (3.7)	3.47 (2.66, 4.51)	<0.001
Brush teeth frequently			
*Yes*	2791 (91.1)	1.38 (1.15, 1.65)	0.001
*No*	191 (6.2)	3.45 (2.80, 4.26)	<0.001
Quantity of toothpaste for brushing teeth			
*Rice grain*	244 (7.9)	1.24 (0.99, 1.54)	0.060
*Pea seed*	1112 (36.3)	1.22 (0.98, 1.52)	0.069
*Full length of bristles*	1534 (50.1)	4.00 (3.27, 4.90)	<0.001
Recently topical fluoride application			
*Yes*	587 (19.2)	1.51 (1.30, 1.77)	<0.001
*No*	2309 (75.4)	2.30 (1.72, 3.08)	<0.001
Need for prosthesis			
*Yes*	1393 (45.5)	0.62 (0.55, 0.69)	<0.001
*No*	1425 (46.5)	10.05 (8.47, 11.94)	<0.001
Consume alcohol			
*Yes*	644 (21.0)	0.92 (0.81, 1.05)	0.202
*No*	2370 (77.4)	5.67 (4.50, 7.15)	<0.001
Alcohol consumption frequency			
*Everyday*	194 (6.3)	0.85 (0.67, 1.08)	0.190
*Sometimes*	443 (14.5)	0.82 (0.66, 1.02)	0.077
*Never*	1963 (64.1)	5.82 (4.73, 7.15)	<0.001
Tobacco use			
*Yes*	317 (10.3)	0.78 (0.65, 0.93)	0.006
*No*	2669 (87.1)	7.76 (5.54, 10.88)	<0.001
Pray			
*Yes*	2454 (80.1)	0.98 (0.85, 1.15)	0.848
*No*	508 (16.5)	4.98 (4.14, 6.00)	<0.001
Religious person			
*Yes*	2495 (81.4)	0.98 (0.85, 1.14)	0.807
*No*	462 (15.1)	5.02 (4.18, 6.02)	<0.001
Spiritual person			
*Yes*	2270 (74.1)	1.11 (0.97, 1.26)	0.117
*No*	677 (22.1)	4.33 (3.65, 5.15)	<0.001
Have beliefs in life			
*Yes*	2560 (83.5)	1.02 (0.86, 1.21)	0.831
*No*	388 (12.7)	4.82 (3.94, 5.91)	<0.001
Prevalence of edentulism			
*Yes*	2200 (71.8)	3.23 (2.85, 5.22)	<0.001
*No*	863 (28.2)	1.70 (1.48, 1.96)	<0.001
Teeth lost due to caries			
*0 teeth*	863 (28.2)	1.72 (1.48, 2.01)	<0.001
*1–10 teeth*	1074 (35.1)	2.00 (1.68, 2.38)	<0.001
*11–20 teeth*	466 (15.2)	1.45 (1.22, 1.73)	<0.001
*21–32 teeth*	660 (21.5)	3.23 (2.85, 3.66)	<0.001
Prevalence of caries experience			
*Yes*	2859 (93.3)	2.07 (1.50, 2.86)	<0.001
*No*	204 (6.7)	2.44 (1.75, 3.38)	<0.001
Prevalence of untreated caries			
*Yes*	1566 (51.1)	3.86 (3.54, 4.20)	<0.001
*No*	1497 (48.9)	1.50 (1.34, 3.38)	<0.001
Participate in social activities			
*Yes*	825 (26.9)	0.98 (0.87, 1.11)	0.807
*No*	2097 (68.4)	5.00 (4.01, 6.22)	<0.001

* *p* value calculated by Qui-square test.

**Table 4 ijerph-20-06661-t004:** Final multivariate-adjusted model for association between dental caries experience and untreated dental decay on OHRQoL in subjects in Paraíba, Brazil.

Independent Variables	*n*	RR (95% IC)	*p*-Value *
Prevalence of untreated caries	1566	1.54 (1.37–1.72)	<0.001
Prevalence of edentulism	2200	1.29 (1.08–1.53)	0.005
Need for prosthesis	1393	0.94 (0.90–0.97)	<0.001
Age Groups			
*12 years-old*	194	1.48 (1.02–2.15)	0.036
*15–19 years-old*	817	1.95 (1.34–2.83)	<0.001
*35–44 years-old*	1302	1.80 (1.23–2.64)	0.002
*65–74 years-old*	750	1.22 (1.09–1.37)	0.001
Sex	3063	1.47 (0.98–2.21)	0.064

RR, rate ration; IC, confidence interval; *p*-value, probability of significance. * Calculated by Qui-square test.

## Data Availability

Due to the nature of the research and legal/ethical issues, supporting data is not available.

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
