# Peer review of "Dental Caries, Tooth Loss and Quality of Life of Individuals Exposed to Social Risk Factors in Northeast Brazil"

_ijerph, 2023, doi:10.3390/ijerph20176661_

Round 1

Reviewer 1 Report

Review of the article entitled ‘ Dental caries, tooth loss and quality of life of individuals ex-2 posed to social risk factors in Northeast Brazil’ ID: ijerph-2537394

This is a good population-based study from Brazil – considering a relevant topic. However I have some questions and remarks that the authors could consider before recommending publication

The introduction section provides background for the present study. In my opinion it is less balanced by being  is heavily focused on caries and tooth loss epidemiology and thus leaving less information regarding the OHRQoL measure which is main outcome of this study.

Is OHIP 14 is the measure used for assessment of OHRQoL and according to the authors this index is validated in the Brazilien population context. However, the study considers a particular subgroup of the Barzilien population and my question s whetjer OHIP 14 has been validated in the population considered in this study. My further question is whether OHIP 14 is a suitable index for the youngest age group investigated . Remember that oral quality of life is age dependent and other indices (as for instance OIDP)  have different versions for children/adolescents and adults.

Caries is measured using DMFT index and also tooth loss. Does the authors see any limitations  connected to the use of this index as a measure of tooth loss?

The sampling strategy is described and seems complex. It should be clear what is the primary sampling unit for this study and whether this unit represents a cluster that should be accounted for in the analyses. If cluster sampling has been employed  how was this adjusted for in the statistical analyses?

Please indicate the justification for the dichotomization of OHIP14 item responses

My  main objection to this study regards the result section. It seems difficult to follow and there has not been identified a primary exposure . Further there is no correspondence between the description of table 2 and the presentation of table 2 in the manuscript. The same yields table 1 and so on.

This makes it difficult to interpret the strategy of analysis utilized as well as the results. Please check and revise. 

Can be improved

Author Response

All responses to each Reviewer 1 query are attached in an attached file. Please see the attachment.

Reviewer 2 Report

The article "Dental caries, tooth loss, and quality of life of individuals exposed to social risk factors in Northeast Brazil" presents a study exploring the relationship between a low socioeconomic level and the incidence of oral diseases in the Northeast region of Brazil. While the work performance is quite sufficient, it lacks fundamental scientific novelty, as the connection between a low socioeconomic status and oral health issues is well-established in existing literature. Nevertheless, the research still holds value in shedding light on the specific context of Northeast Brazil and its potential implications on the quality of life for affected individuals.

The strengths of the article lie in its robust data collection and analysis, providing valuable insights into the oral health status of individuals in the region exposed to social risk factors. The research serves as a valuable contribution to the local health community, as it highlights the severity of oral diseases and their impact on the well-being of the population in the study area.

However, to further enhance the significance and appeal of the article, several recommendations can be made:

Comparison with Other Similar Regions: One way to strengthen the article's significance would be to compare the findings with other regions in Brazil or even in other countries with similar levels of income. By doing so, the authors can assess whether the observed patterns are specific to the Northeast region or more widely prevalent across similar socioeconomic settings. This comparative approach would offer a broader perspective and validate the local findings in a broader context.

Presenting Hypothetical Measures: Given that directly impacting the economic level of the region might be challenging, the authors could hypothetically propose a set of measures or interventions that could potentially improve oral health outcomes in such a context. These measures could include community-based oral health promotion programs, education initiatives, affordable preventive care, and accessible dental services. While these hypothetical proposals may not change the current economic situation, they can offer valuable insights for policymakers and healthcare professionals in addressing the oral health challenges faced by the population.

International Relevance: To increase interest from foreign readers, the authors could draw comparisons with other countries facing similar social risk factors and oral health issues. By incorporating a global perspective, the article becomes more appealing to a wider audience and contributes to the body of knowledge on oral health disparities on a global scale.

In conclusion, while the article "Dental caries, tooth loss, and quality of life of individuals exposed to social risk factors in Northeast Brazil" lacks groundbreaking scientific novelty, it still holds value in its localized context. By incorporating the recommended enhancements - comparisons with similar regions, presenting hypothetical measures, and considering examples from other countries - the article can become more compelling and relevant to a broader audience. Overall, the research contributes to the existing literature and serves as a stepping stone for future studies in this field.

Author Response

All responses to each Reviewer 2 query are attached in an attached file. Please see the attachment.
